# Driving Forces and Socio-Economic Impacts of Low-Flow Events in Central Europe: A Literature Review Using DPSIR Criteria

**Lukas Folkens \*, Daniel Bachmann and Petra Schneider**

Department of Water, Environment, Civil Engineering and Safety, Magdeburg-Stendal University of Applied Sciences, 39114 Magdeburg, Germany; daniel.bachmann@h2.de (D.B.); petra.schneider@h2.de (P.S.)
\* Correspondence: lukas.folkens@h2.de; Tel.: +49-3931-886-4650

**Abstract:** Recent drought events in Europe have highlighted the impact of hydrological drought and low-flow events on society, ecosystems, and the economy. While there are numerous publications about flood risk management and the socio-economic consequences of floods, these have hardly been systematically dealt with in the scientific literature regarding low flows. This paper fills this gap by summarizing the current state of research in the form of a systematic literature review combining the criteria of the drivers–pressures–state–impacts–responses (DPSIR) framework with the propositional inventory method. In particular, the driving forces of low-flow events, their pressures, and the impacts on different economic sectors such as navigation, fisheries, industry, agriculture, forestry, energy, and tourism and recreation as well as resulting competitive usage claims for water and responses are presented and validated through expert interviews. In doing so, the study examines the causal chain of low-flow events and serves as a fundamental base for the future development of a damage cost database for low-flow events by preparing literature data on the socio-economic consequences of low-flow events for parameterization.

**Keywords:** low-flow risk management (LFRM); hydrological drought; DPSIR framework; water security; climate mitigation; water stress; external effects of low-flow events

## 1. Introduction

Ripple et al. (2019) [1], along with more than 11,000 signatory scientists, emphasized that the world is moving toward a climate emergency and that many of the climate change plans based on the Paris Agreement are not ambitious enough to prevent untold human suffering. That same year, Vogel et al. (2019) [2] pointed out that the northern hemisphere heat and drought of the summer of 2018 would most likely not have occurred without anthropogenically accelerated climate change and that such extremes were not observed before 2010. According to the "Global Risks Report 2022" of the World Economic Forum [3], extreme weather events and the failure of climate protection measures are among the greatest global risks for humankind in the short, medium, and long term, accompanied by multi-layered social, ecological, and economic interdependencies. One of these immediate (direct) effects is a low-flow risk that has been increasing for years. The dry summers of 2018 and 2019, for example, caused new low-water extremes in many European streams, in duration, discharge, and water level [4]. Overall, low-flow events in the northern hemisphere are expected to continue to accumulate in the future [5–8]. Model projections under different representative concentration pathway (RCP) scenarios consistently indicate stronger precipitation deficits in the summer months and associated meteorological and hydrological droughts in Europe [9]. In addition to quantitative consequences, this will also affect surface water quality by, e.g., increasing water temperatures, decreasing flows, and enhancing algal blooms [7].

It should be noted that no uniform definition for the term "drought" exists internationally and, moreover, the practicability of such a definition is questioned in the literature [10], as definitions relate to regional specifics. In 1980, Dracup et al. [11] already distinguished between meteorological, hydrological, and agricultural drought. Wilhite and Glantz [12] added the term socio-economic drought in 1985. This classification is generally still used today, which is why it is also adopted in the context of this paper. A meteorological drought describes a temporary, negative, and severe deviation from average precipitation over a significant period in a river basin or region [13]. Hydrologic drought (or blue-water drought) is defined as a significant deficiency of streamflow, groundwater, or reservoir and lake storage [13]. Consequently, low-flow events in rivers are an expression of hydrological drought, which is why this is of particular importance in the context of this study. Further, surface water in general represents one of the most important components for ensuring water security [14,15]. As indicated, agricultural drought (or green-water drought) exists in addition to those already described, but it is only of peripheral importance to this paper. It describes an unusual and significant deficiency of water stored in or on the soil or vegetation [13]. While the three types of drought mentioned above focus in essence on the physical dimension of drought phenomena, socio-economic drought refers to the impact of water shortages on socio-economic systems and thus to the change in supply and demand structures [12,13,16]. In addition to the hydrological aspects, the ecological and socio-economic consequences of low flows are increasingly coming into the focus of public perception, caused by the succession of several drought years in 2018, 2019, and 2022. It must be noted, however, that low-flow events have occurred repeatedly in history. For Central Europe, the German Federal Institute of Hydrology (BfG) offers a summarizing data basis of historical low-flow events with the platform "Undine" [17]. A total of fifteen low-flow events have been recorded for the Elbe since 1893. For the Rhine, the figures go back to 1822 and a total of 22 events have been registered.

This study addresses the causal chain of low-flow events, with special reference to socio-economic consequences by identifying, in the form of a literature review, the driving forces, pressures, state, and socio-economic impacts for various economic sectors such as navigation, fisheries, industry, agriculture, forestry, energy, and tourism and recreation sectors, as well as possible responses. Ecological effects are not primarily included in the analysis but are partially considered within the concept of ecosystem services. These are benefits that people derive from ecosystems, such as rivers. According to the most common classification according to the Millennium Ecosystem Assessment (MEA) of 2005, they can be divided into provisioning, regulating, supporting, and (socio-)cultural services [18]. They are discussed in more detail in the Results section.

To identify the causal chains of low-flow events, the drivers–pressures–state–impacts–responses (DPSIR) framework approach of the European Environment Agency (EEA) from 1999 [19] will be combined with the propositional inventory method, as explained in the Materials and Methods section below. The aim of this approach is, on the one hand, to understand the main drivers of low-water events in terms of socio-economic consequences and, on the other hand, to record and systematize them comprehensively. While there is a long scientific history for flood risk management, e.g., [20–22], and the driving forces and socio-economic consequences of floods [23,24], these have hardly been systematically dealt with in the scientific literature regarding low flows. This paper fills this gap by summarizing the current state of research. In addition, the literature results were validated through guideline-based expert interviews. Overall, the following research questions are addressed:

1.  What are the main drivers of low-flow events?
2.  How can the socio-economic impacts of low flows be classified?
3.  What externalization problems arise from low-flow events?

The study represents a preliminary work for the development of a damage cost database for low flows and was prepared within the research project DryRivers [25] at the University of Applied Sciences Magdeburg-Stendal. Within the joint project, goals, requirements, strategies, and instruments for a sustainable low-flow risk management

(LFRM) will be developed. The damage cost database to be developed can be seen as one component of this and should make it possible in the future to derive generalized and validated damage functions for typical low-flow impacts. The current study will serve as a basis for this.

As the third research question suggests, externalization problems will also be addressed in this paper. An external effect (also externality) occurs when the preference order or the utility situation of an economic entity is influenced by variables that are determined by the activities of other economic entities [26]. Appendix A contains an example scenario that illustrates the possible impact of (negative) externalities using a fictitious river. The example reveals that the decisions of economic agents can affect the preference order of uninvolved third parties or the community. Regarding low-water risk management, the need to harmonize different usage claims as well as ecological and economic concerns is evident. As common goods, river systems are subject to the "tragedy of the commons" theorem described by Ostrom (2008) [27]. Without regulation, in principle no one can be excluded from the use of commons, but at the same time there is rivalry in consuming them. Thus, there is an overuse of common goods. The present study is intended to take this fact into account.

## 2. Materials and Methods

For meta-analysis, the propositional inventory method [28] is used, resulting in a systematic literature review [29]. The method of the propositional inventory is a literary-analytical procedure, which increases the intersubjective comprehensibility through a systematized approach. As with other literary-analytical methods, the goal is the generalization of the research object [28]. To holistically capture the causal relationships of low-flow events, especially their drivers and impacts, the analysis is combined with the DPSIR framework approach [19]. This is a tool for representing environmental pressures and environmental protection measures. The model describes causal relations of influencing variables.

- Driving forces are processes that can exert pressure on the natural and anthropogenic environment (here, e.g., persistent precipitation deficits).
- Pressures are the resulting environmental pressures (here, e.g., low water levels and alteration of stream hydrology).
- State is the condition of socio-economic and environmental compartments subject to pressures (here, e.g., water balance, ecosystem services, or socio-economic usability).
- Impacts are the specific consequences caused by the environmental stress (here, in particular, socio-economic damage due to, e.g., limited inland navigation, reduced water withdrawals, drinking water shortages, etc.).
- Responses are the societal responses to environmental stress (here, e.g., mitigation measures).

To generate the propositional inventory, a keyword search was conducted in Google Scholar, Scopus (both May 2022 with an update in January 2023), and Web of Science (October 2022). This showed that Google Scholar has the highest publication density compared to the other databases, which was also evidenced by Harzing and Alakangas (2016) [30] in a longitudinal and cross-disciplinary comparison. However, it was also indicated that many of the papers found by Google Scholar are so-called "stray citations", where minor variations in referencing result in duplicate entries for the same paper. Since the driving forces and socio-economic impacts as well as responses to low-flow events have been underweighted by the scientific community compared to floods, grey literature was also considered. Only papers in the English language were searched, and no study period was specified. The search terms used in the keyword search are shown in Figure 1.

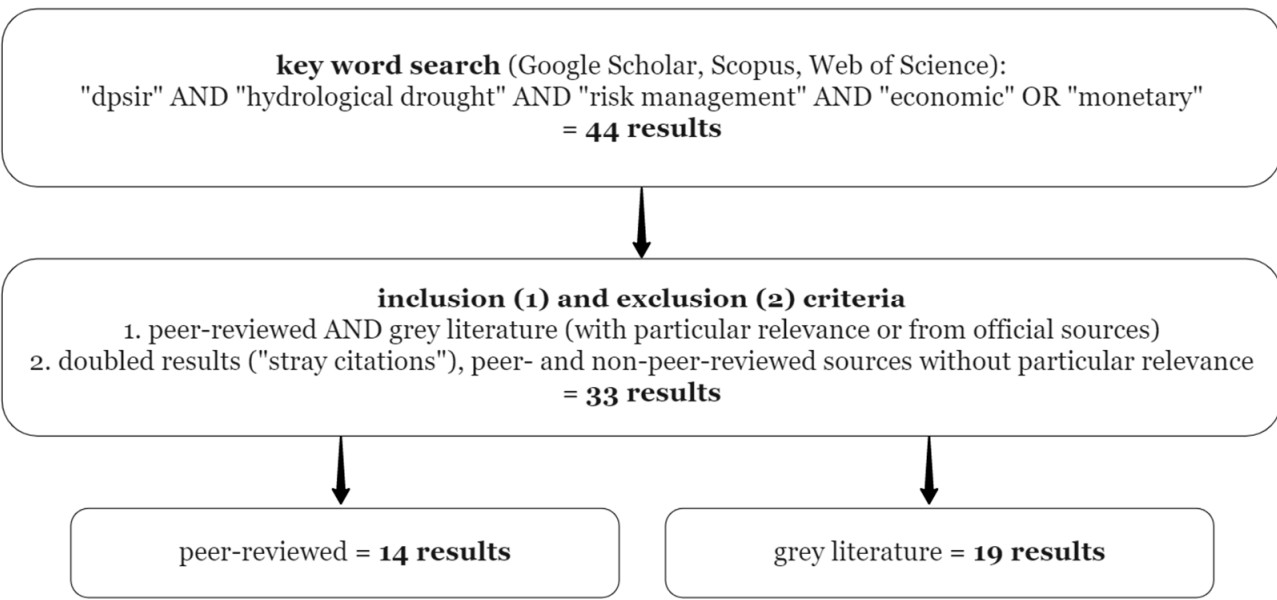

**Figure 1.** Methodological approach (own illustration).

To reflect the research focus described previously, the search terms used are intended to identify existing DPSIR analyses in the topic area of hydrologic droughts that also pay particular attention to economic and risk management aspects. Although relatively few search terms were used, only 44 papers were found in total. The challenge in selecting the search terms was to keep the search query neither too open nor too narrow. This is the main reason why further specifying search terms, such as "low water" OR "low flow" AND "river" OR "stream", were analyzed only in the later course of the investigation (see Section 3). For a critical reflection of the chosen search terms, see Section 4. Starting from the 44 publications found, the next step was to define inclusion and exclusion criteria to pre-filter the literature relevant to the research questions mentioned in Section 1. Articles irrelevant to the topic and stray citations were then sorted out, resulting in 33 remaining articles. This corresponds to 75 percent. In a final step, a distinction was made between peer-reviewed (*n* = 14) and grey literature (*n* = 19), with much of the grey literature being publications by official bodies such as the UN or scientific documents that have not been peer-reviewed but have undergone other review procedures (e.g., master's and doctoral theses). These remaining 33 sources constitute the research subject for the present study. They were evaluated against the research questions described in Section 1, with a semi-automated content analysis using the Adobe Pro search feature. This includes examination of the individual DPSIR components but also more in-depth analysis using the additional search terms described above. For this purpose, the literature was first categorized according to the scheme shown in Figure 2, scanning the entire text of each publication for the corresponding keywords and topics.

What mattered here was if the subjects were covered in a meaningful way, not if the exact term occurred. In addition, a gradation was made by marking whether a topic was covered comprehensively (●) or only partially (◑). Even if the methodological approach has the fundamental goal of objectifying the research topic, it must be pointed out that the classification could not ultimately be made without a subjective weighting. The approach aimed to rank the remaining sources according to their relevance for the specific research topic, with A for high relevance, B for medium relevance, and C for low relevance. As a preliminary assumption, it was determined that in principle only journal articles and monographs can achieve A status, whereby in the case of journal articles it was also important whether they are peer-reviewed or not. To achieve an A score, papers had to further include the additional search terms described above and at least four of the five DPSIR components. Moreover, the last point was applied to the remaining papers as a

benchmark for distinguishing between B and C scores. In addition, the bibliography of publications deemed relevant was searched for further sources (snowball technique) [31].

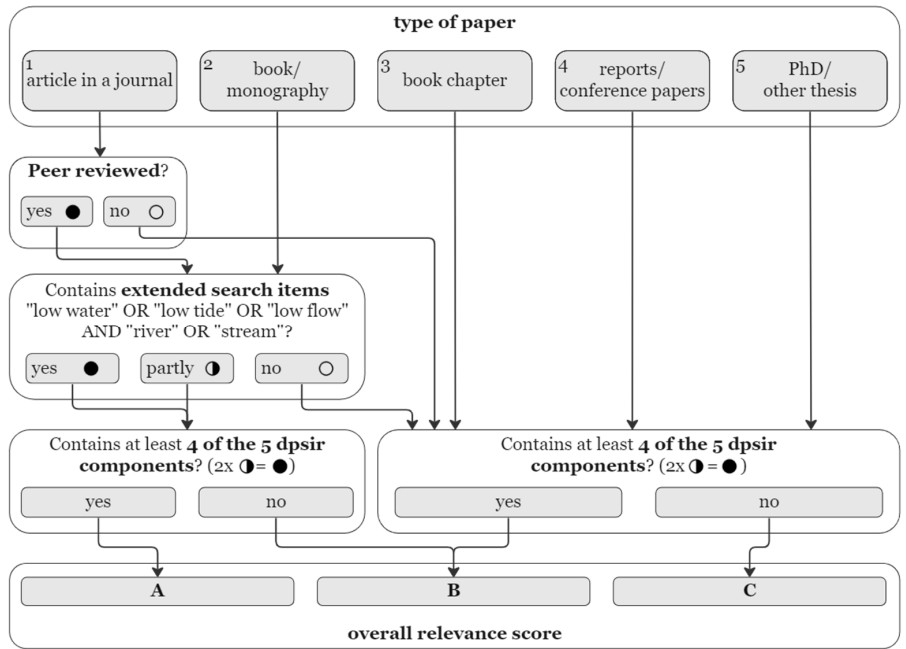

**Figure 2.** Determination of the overall relevance score (own illustration).

　　The literature search itself was not spatially delimited, but a spatial assignment was made afterwards. In contrast, the evaluation of the literature was carried out with special reference to the Central European context and the climatological and socio-economic conditions there. However, relevant information from other parts of the world has been included selectively. More detailed information can be found in the Results section, where, for example, the distinction between agricultural and industrial areas is addressed. The evaluation of the sources was accompanied by an intensive scientific discussion within the research group and the faculty.

　　To validate the results, semi-standardized guideline-based expert interviews, according to Gläser and Laudel (2010) [32], were conducted as well. This is mainly because, as has been indicated, the literature on the research field under investigation is small in quantity. In addition, this approach helps to close the gaps resulting from the restrictive selection of the described search terms. The expert interviews were designed to identify shortcomings in the scientific literature and to confirm existing information. For this purpose, a total of ten experts from different sectors were interviewed, all of them with a practical connection to one or more Central European rivers. For the selection of the interview partners, the associated partners of the DryRivers project were initially consulted. In addition, existing cooperation networks were used. Through this approach, most of the low-flow consequences described by the literature were covered. For a list of the experts interviewed, see Appendix B. The experts were asked about sector-specific low-flow consequences and conflicts in water use. The interviews were recorded and transcribed so that they could be evaluated for this study. For the analysis of the interviews (see Section 3.6), the statements were compared with the literature results.

## 3. Results

### 3.1. Overview on the Bibliographic Results

　　Table 1 lists the 33 papers identified through the keyword search described in Section 2 and the selection process shown in Figure 2.

**Table 1.** Propositional Inventory.

| Publication | Type of Paper | Peer-Reviewed | Spatial Reference | Driver | Pressure | State | (Socio-Economic) Impact | Response | Extended Search Terms * | Overall Relevance |
|---|---|---|---|---|---|---|---|---|---|---|
| Bryan et al., 2020 [33] | 1 | ● | UK | ● | ● | ● | ● | ◑ | ● | A |
| Cap-Net UNDP, 2015 [34] | 2 | ○ | Global | ◑ | ● | ● | ● | ● | ◑ | A |
| Ribbe et al., 2013 [35] | 2 | ○ | Mekong River Basin | ● | ● | ● | ● | ● | ● | A |
| Tsakiris, 2016 [36] | 1 | ● | Global | ● | ● | ● | ● | ● | ◑ | A |
| Wang et al., 2019 [37] | 1 | ● | Heihe River (China) | ● | ● | ● | ◑ | ● | ● | A |
| WWF and GIWP, 2016 [13] | 2 | ○ | Global | ● | ● | ● | ● | ● | ● | A |
| Al Hussain, 2017 [38] | 5 | ○ | Lower Teesta River Basin (Bangladesh) | ◑ | ● | ◑ | ● | ● | ◑ | B |
| Assubayeva, 2022 [39] | 5 | ○ | Central Asia | ◑ | ● | ◑ | ● | ● | ◑ | B |
| Chilikova-Lubomirova et al., 2020 [40] | 4 | ● | Bulgaria | ● | ● | ◑ | ● | ◑ | ◑ | B |
| Chung et al., 2009a [41] | 1 | ● | South Korea | ◑ | ◑ | ◑ | ◑ | ◑ | ◑ | B |
| Chung et al., 2009b [42] | 1 | ● | South Korea | ◑ | ◑ | ○ | ○ | ◑ | ◑ | B |
| Flörke et al., 2011 [43] | 4 | ○ | Europe | ● | ● | ● | ● | ● | ● | B |
| Holman et al., 2021 [44] | 1 | ● | UK | ◑ | ◑ | ● | ◑ | ● | ◑ | B |
| Ilcheva et al., 2015 [45] | 1 | ● | Southeast Europe | ◑ | ◑ | ◑ | ○ | ◑ | ◑ | B |
| Kolcheva et al., 2016 [46] | 1 | ● | Bulgaria | ◑ | ◑ | ◑ | ○ | ◑ | ◑ | B |
| Kossida, 2015 [47] | 5 | ○ | Greece | ● | ● | ● | ● | ● | ● | B |
| Kovar et al., 2009 [48] | 4 | ○ | Czech Republic | ● | ● | ◑ | ◑ | ● | ● | B |
| Olsson et al., 2010 [49] | 4 | ○ | Europe | ◑ | ◑ | ◑ | ● | ● | ● | B |
| WHO, 2011 [50] | 2 | ○ | Europe | ◑ | ◑ | ◑ | ◑ | ● | ◑ | B |
| Vargas Amelin, 2016 [51] | 5 | ○ | Spain | ● | ● | ● | ● | ● | ◑ | B |
| Zucaro et al., 2017 [52] | 1 | ● | Italy | ◑ | ◑ | ● | ◑ | ◑ | ◑ | B |
| Allen-Dumas et al., 2021 [53] | 1 | ● | Global | ◑ | ◑ | ○ | ◑ | ◑ | ○ | C |
| Daoud, 2015 [54] | 5 | ○ | Egypt | ◑ | ◑ | ◑ | ◑ | ◑ | ● | C |
| Eddoughri et al., 2022 [55] | 1 | ● | Morocco | ◑ | ◑ | ○ | ◑ | ○ | ○ | C |
| Mishra et al., 2018 [56] | 3 | ○ | Vietnam | ◑ | ◑ | ○ | ○ | ◑ | ◑ | C |
| Nyangena, 2018 [57] | 5 | ○ | Kenya | ● | ◑ | ◑ | ◑ | ○ | ○ | C |
| Perović et al., 2021 [58] | 1 | ● | Serbia | ◑ | ◑ | ○ | ○ | ○ | ○ | C |
| Pociask-Karteczka et al., 2018 [59] | 3 | ○ | Poland | ◑ | ◑ | ◑ | ◑ | ○ | ○ | C |
| Reckermann et al., 2022 [60] | 1 | ● | Baltic Sea region | ◑ | ◑ | ○ | ○ | ○ | ○ | C |
| Soares et al., 2019 [61] | 1 | ● | Portugal | ○ | ○ | ○ | ○ | ○ | ○ | C |
| Sperotto, 2013 [62] | 5 | ○ | North Adriatic coast | ◑ | ◑ | ◑ | ◑ | ○ | ○ | C |
| Swart et al., 2012 [63] | 4 | ○ | Europe | ◑ | ◑ | ○ | ◑ | ○ | ○ | C |
| Wade et al., 2006 [64] | 4 | ○ | UK | ◑ | ◑ | ◑ | ◑ | ◑ | ◑ | C |
|  | | 15 | | | | | | | | A = 6, B = 17, C = 12 |

● match, ◑ partially covered, ○ no match, 1 article in a journal, 2 book/monography, 3 book chapter, 4 reports and conference papers, 5 PhD and other thesis, A high relevance, B medium relevance, C low relevance, * includes "low water" OR "low flow" AND "river" OR "stream".

These are sorted by relevance according to the relevance scheme described above. Those with the same score were sorted alphabetically. The oldest source is from 2006. Eight publications were written in 2020 or later. The vast majority (42 percent) of publications are peer-reviewed journal articles (*n* = 14). Furthermore, four monographs (12 percent) and two book chapters (6 percent) are listed. A total of six sources (18 percent) can be characterized as reports or conference proceedings. Seven publications (21 percent) are final theses. The majority of these are doctoral theses (*n* = 5). More than half of the literature relates to Europe (*n* = 19). According to the scheme shown in Figure 2, a total of six papers were given a relevance score of A, 17 were given a relevance score of B, and twelve were given a relevance score of C. The distribution of the individual components of the DPSIR framework is discussed further in the following sections.

### 3.2. Driving Forces and Pressures Causing Low-Flow Events

As Figure 3 illustrates, the driving forces of low-flow events can be categorized into four main drivers.

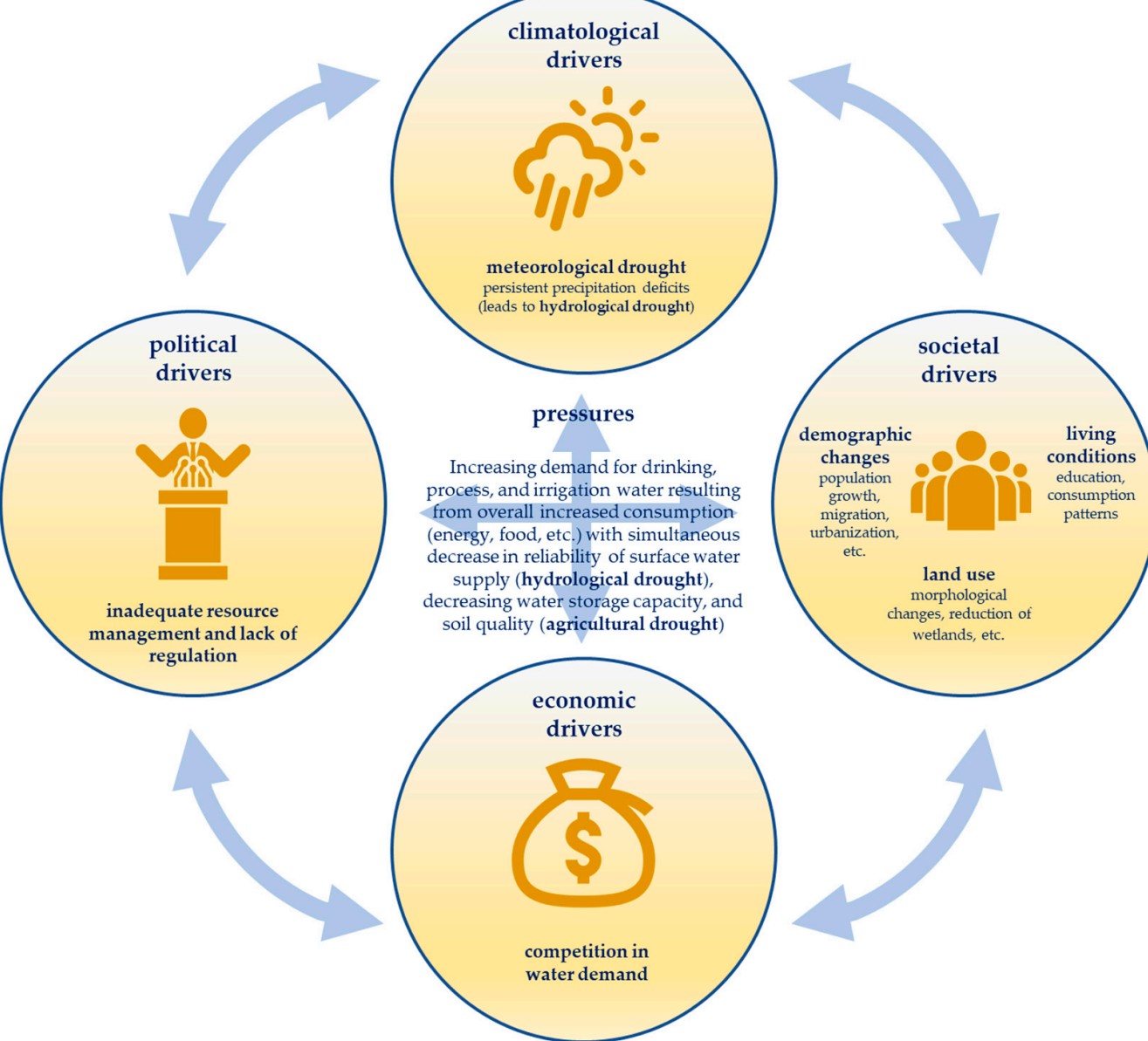

**Figure 3.** Driving forces and pressures causing low-flow events (own illustration).

According to the literature, meteorological drought induced by climatological changes is the fundamental driving force of hydrological and agricultural drought [13,36,52], which subsequently are ecosystem pressures. In this context, precipitation deficits (including snowfall) are ultimately only a partial phenomenon that is additionally accompanied by high temperatures, low relative humidity, more hours of sunshine, less cloud cover, and thus increased evaporation [65]. For the United Kingdom, for example, Bryan et al. (2020) [33] note that the risk of water scarcity due to climate change is a key area for action. Water scarcity describes structural stress conditions due to lack or deficiency of water [36], whereas hydrological drought is a temporary phenomenon leading to water shortages. The term water shortage describes the temporary deficit of water supply to meet the demands [36]. Along with the climatological drivers, human activities affect water supply, release, and storage [37]. These must be included as anthropogenic drivers, as they have a direct impact on the spread of different drought types and may even be the main cause of droughts and low-flow events under certain circumstances [48,66,67]. They can be divided into political, social, and economic drivers. All three can affect streamflow and thus water supply, while interacting with each other. Thus, economic development can put pressure on already fragile systems [37,47]. A distinction must be made here as to whether the economy of the particular region is more agricultural or industrial. For agricultural economic structures, water demand is relatively higher than for technology- and knowledge-intensive industries [68] and changing climate will directly affect water supply and demand patterns [55]. In addition to relatively lower water use, water use efficiency is generally higher in industrial areas [35,37]. In absolute terms, however, it must be stated that economic growth overall increases the demand on vulnerable water resources. Wang et al. (2019) [37] showed this with the example of the Heihe River Basin, where economic but also population growth was identified as a driving force. Population growth is also identified as a key driver by Ribbe et al. (2013) [35] in relation to the Mekong region. As a demographic driver, population growth, just like migration and urbanization, affects vulnerability to hydro-meteorological extremes [35,47] and increases consumption-related pressure on global water resources [13,46]. This includes not only public water supply but also the rising demand for irrigation, process, and cooling water [45] due to the growing demand for food, energy, and industrial products [35,37]. In rural areas, but especially in urban areas, the pressure on vulnerable water resources is increasing, as the processes described accumulate more strongly there [56].

The competing demands raise the risk of low water levels considering external effects, as shown in the example in Appendix A as well. Other important drivers, however, which go along with the ones described so far, are changes in living conditions (e.g., shifting consumption patterns [13,47]) and land use [47,51,63]. Here, there is a high degree of dependence on the economic structures already described. For example, the extension and intensification of agricultural land may be accompanied by the reduction of forest areas and land degradation, or dams may be built for power development and irrigation purposes [38,69–71]. Changes in land use also affect soil structure and infiltration, which in turn affects the amount of surface runoff [58]. As land use change is a major driving force, it must be regulated by political authorities. However, inadequate resource management and regulation in land use and economic terms can accelerate or even cause low-flow trends [13,34,43].

The driving forces and resulting pressures described above should not be viewed in a linear manner but can slowly accumulate over substantial periods [33,34]. Together, they lead to an increase in drought-induced phenomena, such as low-flow events (*state*). This process is illustrated in Figure 4, with socio-economic impacts emerging from hydrological drought [63] and the developments described. These impacts are discussed in Section 3.4. Ecological and other impacts and underlying pressures were added to Figure 4 for completeness, but only the framed subprocess is of particular relevance to this study, as it shows the causal chain of low-flow events in a socio-economic context.

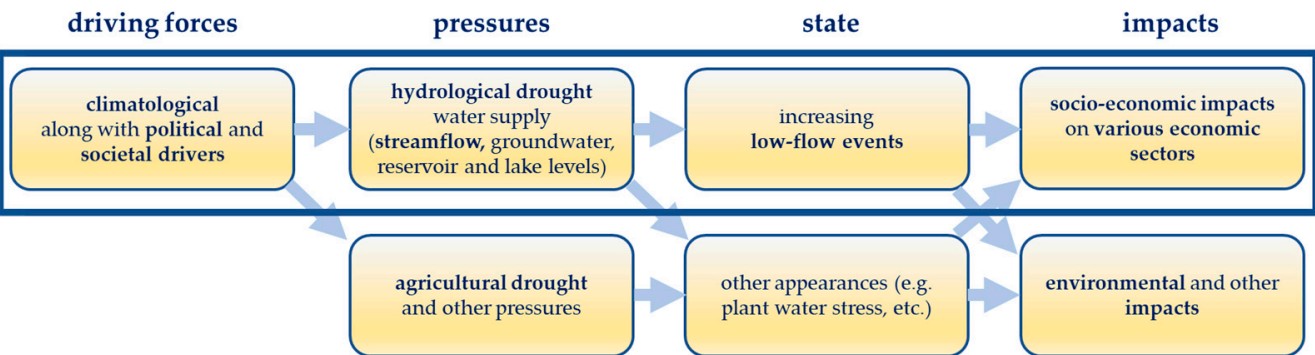

**Figure 4.** From meteorological drought to socio-economic impacts using the DPSIR method (own illustration, partly adapted from Holman et al. [44]).

### 3.3. State

It was already underlined in the Introduction of this paper that there is evidence that low-flow events will increase in the northern hemisphere [5–8]. New extreme values have already been reached in recent years. This is well documented in the literature, so it is not discussed extensively here, but only further references to the literature are provided. For example, Bryan et al. (2020) [33] comprehensively addressed changing conditions in surface watersheds in the United Kingdom. There, with reference to scenario analyses, it is described that the water supplies necessary to meet existing demand could be insufficient in the future during periods of drought, which could lead to water restrictions and temporary withdrawal bans. Wang et al. (2019) [37] also addressed this fact and emphasized that industrialization and urbanization will accelerate this development, by focusing primarily on arid regions in Asia, where desertification trends could intensify. In the present study, however, the focus is predominantly on Central Europe, which is largely characterized by humid areas. Further, melting of glaciers will also have massive impacts on freshwater supplies [70,72,73].

Flörke et al. (2011) [43] as well as Kossida (2015) [47] expected the increase in severe droughts, which will lead to losses of biodiversity, threats to human health, and damage to economic sectors such as agriculture, energy, and tourism particularly in Eastern, Western, and Southern Europe. For Eastern and Western Europe, they cited increasing water withdrawals as one of the main causes, while in Southern Europe, due to climatic changes, water availability is the main problem. Flörke et al. [43] further stated that competition between different water users will occur as a result of the scarcity of water resources, as discussed in more detail in the following sections. This is accompanied by the fact that during the summer months, when water levels are low and will become increasingly low over the next decades, at the same time the demand for water is highest compared to the rest of the year [43].

### 3.4. Socio-Economic Impacts of Low-Flow Events

The socio-economic impacts of low flows are grossly underrepresented compared to those of flood events, both in public perception and in the scientific literature. Unlike floods, whose effects are immediately noticeable, the consequences of hydrological drought build up rather slowly, increase steadily, and depend on regional conditions [43,59,74]. As the following overview (Table 2) illustrates, low-flow events can lead to a variety of socio-economic impacts (socio-economic drought).

**Table 2.** Associated impacts of low flows on various economic sectors.

| Sector | Associated Effects | Pecuniary Effects on Economy and Society |
|---|---|---|
| Inland navigation | Lowering of water levels in rivers and canals makes navigation difficult or impossible [13,47,75] | Reduced transport of goods (in fact due to low fairway depths and induced by, e.g., lack of orders in case of low-flow risk) |
| Tourism and recreation | | Reduced passenger transportation; limited ferry services |
| | Less recreational activities and tourism due to reduction in the amount of water, water level at the surface, and water quality [13,33,49–51] | Losses in the leisure and tourism industry |
| | Reduced runoff and surface water levels for water-related cultural activities | Loss of cultural sites |
| Energy | Reduced flows through hydroelectric power plants or for pumped storage withdrawals [13,39,43,51] | Production losses at hydropower and thermal power plants |
| | Lack of raw materials for energy production(see Inland navigation) | |
| | Reduced discharges and surface water levels for cooling water abstraction [13,43,47,49] | |
| Industry | Reduced runoff and surface water levels for industrial abstraction (service water) | Water risk for industrial users; decline in industrial production and export earnings |
| Water suppliers and households | Reduced surface water levels affect mixing ratio for wastewater discharge | Increased water treatment costs for water supply from bank filtrate; increased costs for wastewater discharge |
| | Reduced runoff, surface water levels, and water quality for domestic and municipal withdrawals [13,33,50,75] | Water scarcity and use restrictions for households and municipalities; losses for water utilities; insufficient water for hygiene purposes; health and well-being effects |
| Public and local government units | Low surface water levels lead to increased administrative burden [36] | Losses due to tax reductions and taxes on hunting and fishing licenses; lack of withdrawal fees from industrial users; administrative costs of issuing and enforcing withdrawal bans (general orders) in low-water events; costs of advertising to reduce water use |
| Aquatic production | Deterioration of aquatic and terrestrial habitats (as a result of increased plant stress, loss of aquatic connectivity, alteration of chemical–biological and hydrological conditions, loss of ecosystem functions) [13,35,36,47] | Reduction of aquatic production (food, medicine, cosmetics, etc.) |
| Agriculture and forestry [1] | | Damage to and reduced growth of crops or crop yields leading to loss of income for farmers and others affected, as well as a decline in food production and simultaneously rising food prices (socio-economic drought); forest losses and forest fires; dairy and livestock losses (due to reduced food and water capacity) |
| | Reduced soil moisture and water for irrigation and livestock supply [33,38–40,44,51] | |
| | Increase in insect infestations, tree and plant diseases as a result of changing ecosystem processes | |

[1] The consequences of agricultural drought are closely interwoven with those of hydrological drought and are therefore included for completeness.

The table basically presents the essence of all the literature reviewed in relation to socio-economic impacts of drought-related events, with Tsakris (2016) [36], WWF and GIWP (2016) [13], Ribbe et al. (2013) [35], and Kossida (2015) [47] in particular providing comprehensive overviews. The impacts presented can be both direct and indirect (stream reference), immediate or delayed (time reference), and tangible or intangible (monetary evaluation) [36]. Furthermore, drought episodes differ in intensity, duration, and spatial extent [34,57,76], which in turn influence the extent of the impacts as well. As the table further illustrates, there are sector overlaps of individual effects. Reducing discharges

and surface water levels in cooling water abstraction, for example, applies equally to the "energy" and "industry" sectors.

Low-flow events develop a high potential for damage, especially where there is a high dependence on surface water supplies [77,78]. When quantifying the economic damage caused by drought, it is striking that estimates in the literature vary widely. For example, Huntingford et al. (2019) [79] estimate the cost of droughts to be USD 1.5 billion globally between 1998 and 2017, representing 33 percent of the costs of weather hazards over the same period. In contrast, van Lanen and Tallaksen (2009) [48] put the impact of the 2003 drought in Europe alone at EUR 8.7 billion. How high the proportion induced by low-flow events is could not be determined in either study. The impact of past droughts on individual European countries and their industries was considered in detail by Kossida (2015) [47].

Some of the listed papers looked at the overall impacts of drought-related events, e.g., in relation to a defined region [35,51,54,59], while others focused on individual aspects of impacts. For example, Bryan et al. (2020) [33] and World Health Organization (WHO) (2011) [50] addressed drought–health linkages, especially in the developed country context, Nyangena (2018) [57] focused on drought-related impacts on pastoral communities in Kenya, and Holman et al. (2021) [44], Chilikova-Lubomirova et al. (2020) [40], and Zucaro et al. (2017) [52] limited themselves to considering agricultural drought impacts and responses, Holman et al. [44] using the United Kingdom, Chilikova-Lubomirova et al. [40] using Bulgaria, and Zucaro et al. [52] using Italy as a case study. Furthermore, the impacts can be categorized into the ecosystem service classifications described in Section 1. This approach is followed, for instance, by a UNESCO study [13] and by Olsson et al. (2010) [49]. According to them, the supply of aquatic products or drinking, irrigation, process, and cooling water by surface water bodies such as rivers or streams can be understood as a provisioning ecosystem service. Water absorption, storage, and release can be cited as examples of regulating services, while nutrient cycling and maintenance of genetic diversity work as supporting services. Cultural services result from interaction with the natural environment, e.g., in the form of education, aesthetics, or recreation.

Specifically, Tsakiris (2016) [36] and Cap-Net UNDP (2015) [34] addressed socio-economic drought (or drought risk) in terms of vulnerability. This is defined by the ability of a system to withstand the exerted pressures [36], and can be affected by variable parameters, such the size of the population, per capita water availability, water use trends, policies, technology, etc. [34]. Depending on intensity, duration, and spatial extent low-flow events can cause damage to production and natural, modified, or human systems [36]. In this context, different escalation levels can be identified, which in turn can lead to cascading effects. Possible phenomena of a first escalation stage would be [13]:

- competitive usage claims,
- small-scale and large-scale disputes or conflicts between water users,
- public discontent and increased social injustice,
- unemployment due to decline in tourism, industrial production, fisheries, and agriculture,
- insolvencies and migration of businesses,
- loss of livelihoods, stoking of fears for the future, and migration,
- increased importation of food, which also means higher food costs,
- food shortages and famine,
- health risks associated with the increase in the concentration of pollutants and the disruption of water and food supplies,
- loss of livelihood for subsistence fishers and farmers,
- pressure on financial institutions (more risks in lending, decrease in capital, etc.),
- damage to land and property and threats to public safety.

These impacts can in part reinforce each other and thus have a cumulative effect. If a low-flow situation persists for several years or even perpetually, even more severe consequences may result. Such a second stage of escalation could be characterized by the following events, for example:

- slowdown in economic development, loss of national economic growth;

- social rebellions and political conflicts, including over water and food;
- dehydration and related health problems and diseases, including death;
- malnutrition and related health problems and diseases, including death;
- worsening social inequality.

In the following subsections, individual impacts are addressed separately, focusing on a subsequent parameterization for developing the damage cost database described in Section 1.

### 3.4.1. Impact on Inland Navigation

The monetary impact of low flows on inland navigation is obvious. When water levels are low, cargo ships can only operate with a part of their maximum capacity, which in turn means that multiple vessels must be used for the same cargo volume, leading to increased costs [13,47,75]. The lower the water level, the less cost-effective the transport of goods becomes. Koetse and Rietvald (2009) [80] estimated that the assumed loss for the Rhine navigation in the drought year 2003 amounts to approximately EUR 91 million. A study published in 2020 calculated economic losses for inland shipping and industry in Germany and the Netherlands during the low-flow period in 2018, totaling EUR 2.7 billion [81]. The report stated that most of the losses (EUR 2.2 billion) resulted from production losses or restrictions in industry, whose raw materials are mainly supplied via waterways, especially the Rhine. As Germany's most important waterway, the Rhine was recently affected by low flows in 2018, 2019, and 2022. But also the Elbe, as another important waterway, had water levels of less than 1.40 m on 240 days in 2018, which is why no navigation could take place on these days [82]. Particularly in the lower reaches of the Elbe, such conditions are also projected for the future, which will make it impossible to use the Elbe as a waterway economically in the long term [43]. Economic efficiency is the decisive dimension in the context of inland navigation, as there is no officially ordered blockage of shipping during low flows in most European countries. The barge operators therefore decide whether and to what extent they use their ships. Due to the shortage of available free capacity of shipping space, they levy so-called "low-water surcharges". The lower the water level, the higher the surcharge for the customers. Depending on the specific conditions of each river, no inland navigation can take place beyond a certain water level. In addition, at low water levels, the waterway narrows, which in turn induces decreasing travel speeds and increasing fuel consumption. As stated in the previous section, socio-economic impacts can be both direct and indirect, immediate or delayed, and tangible or intangible [36]. This can be illustrated by the example of inland shipping. In addition to the short-term additional costs for the use of larger fleets or rerouted traffic (e.g., direct, immediate, and tangible), there is also a risk of induced long-term impacts such as the loss of orders, the relocation of production facilities, or a decline in the attractiveness of the location (e.g., indirect, delayed, tangible, and intangible). In addition, employment relationships are directly and indirectly affected by inland navigation. For the Elbe, a study is available [83] which shows that commercial shipping on the Elbe has regional economic effects on a total of 16,400 direct, indirect, and induced employees. In this context, it is worth attaching the term water risk, which refers to the probability that an entity will experience a harmful water-related event [84]. This risk is perceived differently by each organization and is therefore also evaluated differently.

### 3.4.2. Impact on Tourism and Recreation

Bryan et al. (2020) [33] provided an overview of drought-related impacts on sports and recreation. They explain that low water levels limit water-based recreational activities, such as kayaking, canoeing, swimming, and paddle boarding. This not only has economic consequences for corresponding rental stations and the tourism industry overall, but also reduces physical activity and thus health, which in turn can lead to rising health care costs. In addition, the WHO (2011) [50] points out that the quality of bathing water decreases with lower water levels. Due to the reduced dilution [49], the relative concentration of contaminants accumulates. Furthermore, warmer water temperature, better light pene-

tration, and increased plant nutrient concentrations promote the growth of algae, such as blue-green algal blooms [13,50]. This has a negative impact on the health of humans, as well as fish populations (see Section 3.4.6).

Analogous to the explanations in the previous section, the reduced passenger transport or limited ferry services can be mentioned as another impact to the tourism and recreation sectors [13]. But in addition to the tangible impacts, there are intangible consequences such as the loss of local recreation or education sites and restorative benefits [51]. These cultural ecosystem services result from interaction with the natural environment [18]. They are generally more difficult to quantify than the impacts that can be measured in monetary terms.

### 3.4.3. Impacts on Industry and Energy Production

For industrial and energy companies, streams represent both a withdrawal and an intake medium since they both withdraw water for operational purposes and operate as dischargers. From this perspective, there is competition for use between different water users and the quality of the medium is influenced by its use. For example, the discharge of wastewater into a water body may compete with its use for production purposes, drinking water use, fish farming, recreation, or as a habitat for plants and animals. The economic impacts of increased water withdrawals (e.g., by agricultural users), particularly during periods of drought, were addressed, for instance, by Chakravorty and Fischer (2005) [85]. Insofar as the parties involved do not sufficiently include the impacts associated with the use of the water bodies in their economic decision making, these are referred to as external effects or externalities [26]. The negative effects valued in monetary terms are referred to as external costs. A characteristic feature of external costs is the fact that it is not the polluters who bear these costs but parties who have no direct or indirect market relationship to the activity causing them, or society as a whole. The result is a situation in which river systems are stressed beyond an economically optimal level. The example of withdrawals and discharges from industrial and energy companies is a good illustration of this, which is why it has been taken up at this point. The operating principle of externalities is also explained using the example in Appendix A.

On the energy sector, a prolonged period of low flows can have multi-layered effects. A shortage of cooling water, for example, leads to thermal power plants having to reduce electricity production [13,75], as was observed in France in recent years [47]. In addition to water quantity, water temperature also plays a crucial role, as this has a decisive influence on the efficiency of the plants [86,87]. Along with decreasing efficiency, water withdrawal and discharge can also be regulated through legal restrictions if thresholds are exceeded and ecological damage is caused by the discharged heated water [43,47,49]. For Germany, the Federal Statistical Office [88] presented figures that make clear that most of the commercially used water (84.7 percent), with the majority coming from surface waters, is used for cooling production and power generation facilities. Other process water for production purposes takes second place with 10.7 percent, while water withdrawals for irrigation or workforce purposes or water that goes directly into products accounts for an insignificant share.

In connection with the impacts described in Section 3.4.1, there is also the fact that the decreasing navigation of inland rivers is increasingly compromising the supply of (e.g., energy) raw materials such as coal. But renewable forms of energy generation can also be affected. One obvious example is the vulnerability of hydropower plants to low-flow events [13,39,43,51]. More abstractly, in the context of an agricultural drought, biofuel crop yields may decline [43]. Altogether, the effects described reduced energy security and industrial production, leading to rising prices for industrial goods and energy in the affected regions.

### 3.4.4. Impact on Water Suppliers and Households

Water suppliers and households in watersheds that rely on surface water flows are potential early recipients of low-flow impacts [33,47]. The most obvious and serious

consequence is the limited supply of fresh water, e.g., for drinking or hygiene purposes, which in turn affects health and well-being [13,33,50,75]. In addition to the quantitative deterioration of the water body, low water levels also result in reduced water quality [89], as mentioned above. This condition can be further exacerbated by, for instance, industrial uses during low-flow periods, as shown in the example in Appendix A. Restrictions on use for households and communities may result [13]. Further, reduced runoff can damage infrastructures such as water supply systems especially when water quality is poor or infrastructure is outdated [39,90]. Other associated impacts include lack of water for gardening or domestic swimming pools [47].

### 3.4.5. Impact on Public and Local Government Units

The consequences of low flows on public and local government units can be referred to as secondary, indirect impacts. They include losses due to tax reductions and taxes on hunting and fishing licenses [36], lack of withdrawal fees from industrial users, administrative costs of issuing and enforcing withdrawal bans, or costs of advertising to reduce domestic water use.

### 3.4.6. Impact on Aquatic Production

The fishing industry is one of the sectors particularly affected by low flows in rivers. Low water levels and a degradation in water quality can lead to a decline in fish populations, especially for migratory species [35,47], which in turn can have a significant socio-economic impact through reduced catch rates and lower revenues for fisheries and other industries, such as medical and cosmetics sectors [13,35,36]. Economic impacts in a broader context involve the value chain and the local economy, e.g., fishing equipment manufacturers.

### 3.4.7. Impact on Agriculture and Forestry

As explained in the Introduction, the specific impacts of agricultural drought will only be dealt with here in marginal terms. The discussion is therefore limited to the part of agricultural production that depends on irrigation with water from rivers. Farmers in watersheds that rely on surface water flows are potential early recipients of low-flow impacts [33,38,39,51]. In agricultural areas, this sector of the economy incurs the highest costs compared to the other sectors, e.g., through damage to and reduced growth of crops or crop yields, as Ribbe et al. (2013) [35] showed in the Mekong River Basin. But even in industrialized countries, the effects can be immense, as Flörke et al. (2011) [43] showed for Europe and Holman et al. (2021) [44] highlighted specifically for the United Kingdom. Lack of crop yields and shortage of water resources also affect livestock [38–40,44] and pastoralism [39,57], while leading to rising food prices (socio-economic drought). To counter low flow trends, farmers need to invest in water storage facilities or alternative irrigation equipment, which also increases production costs and thus in turn food prices [40]. Different key figures can be determined for the monetarization of the socio-economic dimension. Water-limited crop productivity measures the damage caused in terms of reduced yields. Irrigation water demand, in turn, can be used to measure the increase in water required for irrigation [91]. Lastly, forest loss and forest fires can also be cited as impacts, as Kossida (2015) [47] showed for different European countries.

### 3.5. Response

As mentioned in Section 3.3, most European Union (EU) member states need to prepare for more frequent drought and low-flow events [43]. While a European Council (EC) Directive was issued in 2007 for the assessment and management of flood risks [92] including the request for standardized flood hazards, risk maps, and risk mitigation plans, a comparable counterpart for drought or low-flow events does not exist on the EU level yet. The Water Framework Directive adopted in 2000 applies to these but does not comprehensively address the individual specifics of low-flow events. Efforts in this area have just been intensified in recent years, both on the regulatory and the scientific side. In

principle, mitigation measures for low-flow events range from an intervention in the natural water cycle, e.g., storage measures, measures within the river, e.g., navigation channel excavation, to measures affecting the impact side, e.g., reducing the water demand. The literature listed in Table 1 presents diverse approaches that can be used either to prevent or mitigate the impacts described. Some address individual industries, while others focus on increasing low-water trends holistically. The implementation of closed water loops for cooling and process water in industrial and energy companies, for instance, can be mentioned as a sector-specific measure that make companies less dependent on water availability [37,43]. Here can also be mentioned the regulation of withdrawal prices, which could be set depending on spatial and temporal conditions [37]. In terms of agricultural consequences, Speretto (2013) [62] asked whether farmers should reduce their livestock numbers in the face of a predicted drought, to cite yet another example. Furthermore, interventions can be differentiated according to whether they are short term in the form of reactive measures or long term and preventive. Examples of reactive measures include use of alternative water sources, relief, or suspension of water use permits in watersheds with low water levels. Preventive measures, on the other hand, describe structural mitigation of droughts [34]. Higher-level approaches such as integrated water resource management (IWRM) [37,48,49,93], drought risk management frameworks [13,34,47], early warning systems [53], or national drought management plans [39,51,76] have been proposed in the literature for this purpose. UNISDR (2009) [94], for example, provided a definition of drought risk management, which describes a systematic process to prevent, mitigate, and prepare for the adverse impacts of drought and related disasters.

Low-flow risk management (LFRM), in particular, is not addressed by any of the papers listed in Table 1, highlighting the need for research in this area. For Europe, initial conceptual frameworks [95] and low-flow risk maps [96] have been developed in recent years. In addition, Hall and Leng (2019) [97] defined the statistical properties of an event and its associated consequences as the fundamental elements of a quantitative low-flow risk analysis. Bachmair et al. (2017) [98] also addressed these basic elements in relation to drought impact functions. In the field of practical application, LAWA (2007) presented the first basic features of an LFRM in Germany [99]. The "Nationaal Watermodel" [100] of the Netherlands is a first comprehensive, model-based tool for a national LFRM. It focuses on the economic consequences of a low-flow event. Nevertheless, practical applicable tools and approaches in the field of LFRM are lacking, especially in the regional context. As has been shown, the impacts are often complex and cross-sectoral, which also implies the need for a cross-sectoral management approach and broad stakeholder participation [34,41]. This is necessary both to identify the most vulnerable at-risk groups and sectors [34] and to increase allocation efficiency along the upper, middle, and lower reaches of river basins and different administrative regions [37]. Comprehensive care must be taken to ensure that not only are competing economic sectors served but also that environmental discharges and the need for healthy freshwater ecosystems are included with appropriate prioritization [45,101]. One approach that examines these complex interactions is the water–energy–food nexus or, by extension, the water–energy–food–ecosystem nexus [102]. To ensure the integral requirements for water, food, and energy security across sectors and in a sustainable manner, the nexus dialogue [103] was developed in this framework as a negotiation and participation format. This can form the basis for a structured implementation of an accepted transformation process in watershed management. Furthermore, the internalization of external effects must also be considered against the background of a resource-appropriate allocation of the scarce freshwater resource. This is not addressed by most sources in Table 1, except for Assubayeva (2022) [39]. He described, referring to Molle (2009) [104], that efforts of the French Water Agencies (Agencies de l'Eau) to address water quality problems by introducing the polluter-pays principle to internalize negative externalities influenced the development of the European Water Framework Directive. Against the background of the internalization of externalities, it seems particularly challenging that rivers rarely extend only within one economic or legal area, but cross borders. Thus, as Holman et al.

(2021) [44] stated, drought-related events are also large-scale regional phenomena that span national boundaries, exacerbating the systemic complexity of both impacts and responses. Granit et al. (2012) [105] as well as Steinmann and Winkler (2019) [106] pointed out the challenges and externalities associated with transboundary water management as upstream water users impact downstream regions, e.g., through agricultural and industrial pollution. For this purpose, Endres (2013) [26] proposed the division of first-, second- and third-order externalities. Externalities of the first category describe the already frequently mentioned direct effects between individual economic entities, while cross-border effects between states can be described as externalities of the second category. In addition to this spatial view of the term, Endres adds a third, temporal dimension. According to this, externalities of the third category describe those effects which the present generation exerts on future generations [26]. From this point of view, the concept fits in well with the guiding principle of sustainable development. The challenges described above are addressed by the DryRivers research project [25], in the context of which the present study was carried out.

### 3.6. Validation of Study Results

To validate the literature findings, expert interviews on the sector-specific low-flow consequences were conducted and analyzed (see Section 2). Respondents were first asked to assign themselves to the sectors listed in Table 2, with multiple responses possible. As can be seen in Table 3, the distribution was balanced, and no sector was left out.

**Table 3.** Representation of sectors according to experts' self-assessment.

| Sector | Representation (Frequency) [1] |
|---|---|
| Inland navigation | 2 |
| Tourism and recreation | 2 |
| Energy and industry | 2 |
| Water suppliers and households | 3 |
| Public and local government units | 3 |
| Aquatic production | 1 |
| Agriculture and forestry | 2 |
| Others | 2 |

[1] Multiple answers were possible.

Responses were cross-referenced with the literature findings described in Section 3. Here, mainly additive information is supplemented, thus confirming statements are only briefly mentioned. Interviews are referenced below by interview number (IN) according to the list of experts in Appendix B.

Regarding inland navigation, both the short- and long-term effects described in Section 3.4.1 were confirmed by experts (INs 1 and 7). It was highlighted that the shipped goods could not easily be absorbed by other modes of transport (road or rail) in terms of volume (IN 1). The lack of reliability as a mode of transport was especially confirmed for the Elbe (IN 1). Furthermore, a distinction must be made between free-flowing rivers and regulated rivers (e.g., by barrages). In this context, it was pointed out that shipping is partly dependent on flow regulation, since largely free-flowing rivers (such as the Elbe River) are more suitable for special transports and non-scheduled cargoes and less suitable for scheduled cargoes (IN 7). It also plays a decisive role for fisheries as well as industrial use whether the river is dam regulated and thus has a guaranteed minimum flow (INs 4, 9, and 10). For the Rur River, for example, it was indicated that the dam system could hold water for 2 years and provide a regulated minimum release (IN 4), thus avoiding extreme low-flow events and ensuring year-round industrial use (INs 4 and 10). The need for industrial water withdrawals, as described in Section 3.4.3, was discussed using the example of the pulp and paper industry along the Rur. The industry, for instance, produces essential hygiene articles and packaging material for medical products and provides employment for many people in the region (INs 4 and 10). For ensuring the minimum water discharge ($5 \text{ m}^3/\text{s}$), they pay fees to the dam operators (INs 4 and 10). For the example of the paper

industry, it was stated that approx. 90 percent of the water used is discharged back into the Rur (IN 10). Regarding industrial discharges, salt discharges were mentioned particularly as an example, which have a negative impact on aquatic life but also on recreational uses due to the accumulating effect in low-flow phases (IN 8).

The monetary consequences of low flows for hydropower energy production described in Section 3.4.3 could be confirmed (IN 4), also pointing out the adverse effects of such cross-barrier structures on fish (IN 8).

It was also confirmed by the experts that low-flow events could influence tourism and recreation (e.g., sport boating, kayaking, canoeing, swimming, recreational value) (INs 1, 5, 7, and 10). Indirect effects, e.g., on bicycle tourism (IN 8) and the accommodation industry (decrease in overnight stays) (IN 3), were added to those mentioned in Section 3.4.2.

In Section 3.4.4, the reduced water quality during low-flow periods was described in relation to the municipal water supply, which was confirmed by the expert interviews (IN 9). For water suppliers, this results in rising treatment costs, which indirectly affect the price of water for consumers (IN 9). A distinction must be made here between direct water extraction, extraction from bank filtrate, and water supply via groundwater recharge (IN 9). Again, the influence of dam control was added as a supplement to this topic (INs 9 and 10). It should also be mentioned at this point that drinking water supply is considered a priority over the other forms of water use by most of the experts interviewed.

The interviews provided additional information on the impacts on administrative units described in Section 3.4.5 as well. For example, the costs of operating and maintaining hydrological monitoring stations must also be considered, since the data collected are needed, for instance, to issue withdrawal bans during a low-flow event (IN 6). Furthermore, the dimensioning of sewer networks and retention basins was addressed, which has to be calculated differently in case of an accumulation of low-flow events (IN 10).

The consequences for the fishery sector, described in Section 3.4.6, could be supplemented by the interviews (IN 8). It was described that low-flow impacts are significantly related to water body size (INs 5 and 8), with increasing siltation being a particular problem in small water bodies (IN 8). Costs result from emergency fish removals and restocking (IN 8).

In the context of agricultural and forestry use of rivers (Section 3.4.7), direct abstraction from the watercourse was addressed (INs 5, 6, and 10) but, in addition, groundwater coupling in groundwater-dominated rivers was also considered (INs 2 and 8). Specifically, the decrease in soil moisture in forest areas induced by low flows was mentioned, which supports the statements in Section 3.4.7 regarding the increase in forest fires (IN 2).

In addition, the experts pointed out that many use claims arise from environmental–economic tradeoffs, particularly with respect to the implementation of the Water Framework Directive (INs 3 and 5). It was confirmed that ecological components such as species composition influence water quality and thus have an indirect economic effect on all forms of use (INs 2, 3, and 5). The relevance of riverine floodplains was addressed by several experts in this context (INs 2 and 7). Furthermore, the interviews made it possible to identify additional stakeholder groups that were not specifically addressed in the literature review. These include the dam operators mentioned (IN 10) but also environmental protection agencies (INs 1, 2, and 7).

## 4. Discussion

A literature review claims to give a comprehensive representation of the state of scientific knowledge at a particular point in time. There are methodological limits to this. The combination of the methods presented, and the inclusion of the DPSIR compartments, meant that only papers that also referenced this methodology were included accordingly. Literature that also deals with driving forces, their pressures, and socio-economic impacts of low-flow events but does not reference the DPSIR approach was therefore filtered out by the search terms. This is a limitation, which, however, is due to the main focus as well as the research questions of the study. In addition to the analysis of socio-economic impacts

as a basis for the development of a damage cost database, another main focus was on the integrated presentation of the causal chain of low-flow events (e.g., Figure 4), also including driving forces, and pressures but also possible responses. Understanding the causal chain is in this relation essential for the development of a holistic low-flow risk management (LFRM), of which the database to be developed should be seen as a component. Given this objective, the inclusion of DPSIR was deemed justified. Since there is a possibility that potential consequences due to the restrictive selection of search terms are not considered, ten expert interviews were conducted to fill this gap. Expanding the survey to include more experts might have revealed further findings. Also, the selection of the experts interviewed was also ultimately subjective, although care was taken to ensure that all the sectors described were adequately covered and all experts had a practical connection to one or more Central European rivers. In addition to the literature search, methodological limitations also arise in the context of post-selection of the researched papers. Even if the scheme shown in Figure 2 attempts to present the relevance assessment in a comprehensible way, a subjective assessment cannot be eliminated in the end. The same applies to the literature analysis itself and consequently to the propositional inventory presented as well as for the interview evaluation. Further, it must be noted that the study provides a qualitative rather than a quantitative description. This is because it is to be considered as preliminary work for the development of a damage cost database, as a component of an LFRM approach. The socio-economic impacts of low-flow events outlined in this study will be parameterized for this purpose. For the pilot areas Middle Elbe, Rur, and Selke, they are prioritized and quantified according to their relevance for the river basin and its stakeholders. The following categories can be used for orientation, whereby the affected groups of people are to be presented in further subdivisions in each case:

- immediate impact on the individual's benefit,
- impairment of the production of goods and services,
- impairments that cannot be attributed but impose costs on the economy.

The resulting monetary loss of benefit is quantified, depending on the stakeholder group, using suitable valuation methods [42]. With respect to the use of industrial and cooling water, the production costs could be modeled under low-water conditions, while a classical market price method is appropriate for fisheries. The latter could quantify the lost profits (fewer fish caught) due to low flows using the prevailing market prices for those fish. For tourism, recreation, and leisure, in turn, willingness to pay (WTP) can be determined by revealed preference (RP) and stated preference (SP) methods [42]. Common approaches for SP are the contingent valuation method (CVM) or choice experiments (CEs). Examples of RP methods include the travel cost approach, as well as hedonic property value and hedonic wage models, or the household production model. The damage costs thus collected are translated into damage models in the next step. An indicator for inland navigation could be, depending on the respective flow conditions of each river, the minimum flow [62] or the minimum fairway depth at which it is still possible to operate navigation. For this purpose, it is necessary to relate the water level as a reference with associated costs, where the lower the water level, the higher the operating costs for individual entities.

As the previous explanations make clear, one of the key points of the database development will be the weighted consideration of the user claims, especially those of the most vulnerable sectors [50]. As an example of a possible user hierarchy, the regulations of the state government of Minnesota (USA) [107] can be mentioned:

1. domestic water supply,
2. uses consuming less than 37.85 m$^3$ of water per day,
3. agricultural irrigation and processing of agricultural products,
4. power production,
5. other uses,
6. non-essential uses (watering lawns, washing cars, irrigating golf courses) [13].

Here, in the event of water scarcity, water users are gradually excluded from water withdrawal according to their prioritization level [107]. It must also be taken into account that public facilities, such as hospitals or nursing homes, but also certain population groups require an uninterrupted water supply for medical reasons (e.g., for dialysis machines) and must therefore be given special consideration [33,50]. In addition to the prioritization aspect, targeted LFRM also requires the development of appropriate communication strategies, since, for example, restrictions on use must be communicated appropriately [50].

As a final point of discussion, it should be revisited that only temperate climate zones in general and Central Europe in particular were referred to. Low-flow consequences for semi-arid and arid areas, which are potentially more serious than in humid areas, were only marginally considered in this study. Basically, it can be stated that the results of the study are transferable to all regions that can be compared with Central Europe both climatologically (temperate zone) and socio-economically (developed world).

## 5. Conclusions

By combining a systematic literature review with the compartments of the DPSIR approach, key drivers, pressures, impacts, and responses related to low-flow events could be analyzed in this study, contributing to an understanding of the causal chains of low-flow events. The literature elicited by key terms was systematized using a propositional inventory and then evaluated with respect to pre-defined research questions. Four areas that act as drivers of low-flow events and lead to pressures could be identified, namely climatological, political, economic, and social factors.

The main focus was on the systematic description of the socio-economic impacts of low-flow events. A comprehensive overview has been prepared for this purpose, presenting multiple consequences for different stakeholders. Based on the reviewed literature, consequences were described for inland navigation, tourism, leisure and recreation, energy and industry, water suppliers and households, governmental units, aquatic production, and agriculture and forestry. Measures to address the intensifying low-flow trends were also listed. The literature findings were subsequently validated through semi-standardized guideline-based expert interviews, confirming most results but also supplementing additional points.

The entire consideration was set in the context of developing a database for low-water-related damage costs, which is intended to support a holistic LFRM. In the process, essential findings were obtained on the requirements of such a database and the areas it should cover. Furthermore, the first parameters for this database could be derived from the described impacts.

As has been shown, the impacts are often complex and cross-sectoral, which also implies the need for a cross-sectoral management and broad stakeholder participation, especially against the backdrop of competing usage claims and the vulnerability of different water users. For this purpose, for example, the water–energy–food–ecosystem nexus and particularly the nexus dialogue were taken up as a negotiation and participation format that can form the basis for a structured implementation of an accepted transformation process in watershed management. A dialogue with relevant stakeholders has already been initiated in part by the interviews conducted. In view of the predicted increase in low-flow events in the future, the internalization of external effects must be considered for the efficient allocation of water resources, especially because rivers extend over multiple administrative regions.

**Author Contributions:** Conceptualization, L.F.; methodology, L.F. and P.S.; software, D.B. and L.F.; validation, L.F.; formal analysis, L.F. and P.S.; investigation, L.F.; writing—original draft preparation, L.F.; writing—review and editing, L.F., P.S. and D.B.; visualization, L.F.; supervision, D.B. and P.S.; project administration, D.B.; funding acquisition, D.B. All authors have read and agreed to the published version of the manuscript.

**Funding:** This research was conducted as part of the DryRivers project, which was funded within the Water Extreme Events (WaX) grant by the Federal Ministry of Education and Research (BMBF) (FKZ 02WEE1628A). The APC was funded through the guest editor P.S.

**Institutional Review Board Statement:** Not applicable.

**Informed Consent Statement:** Written informed consent has been obtained from the participants to publish this paper.

**Data Availability Statement:** Not applicable.

**Acknowledgments:** Gratitude goes to Tim Franke (RWTH Aachen University), who supported the authors both in the literature search in Web of Science and in the preparation of the expert interviews. In addition, the authors would like to thank all those interviewed during the expert interviews for their time and expertise.

**Conflicts of Interest:** The authors declare no conflict of interest. The funders had no role in the design of the study; in the collection, analyses, or interpretation of data; in the writing of the manuscript; or in the decision to publish the results.

## Appendix A

The water level of a river has successively dropped as a result of a prolonged dry period. Commercial enterprise A is a river riparian and has always used the river for the abstraction of cooling water. Since no mechanism for low-flow risk management (LFRM) has been established by the regulatory authorities, the company continues this practice during the low-flow period, as it would otherwise have to obtain cooling water elsewhere at higher costs. This economic decision causes the overall situation to deteriorate and the water level to drop further. On the one hand, this results in the migration possibilities for fish being further restricted and, on the other hand, the combination of reduced flow velocities and (due to the smaller water body) high water temperatures induces an extreme deterioration in the oxygen supply of cold-blooded aquatic animals (fish and invertebrate fauna). As a result, fish die off. Company B, which is also located on the river, is a fishing company that now has massive economic damage. Company C, a small company in the water sports and recreation sector, also has to stop its business activities because it is prohibited to use the water body at such low water levels.

## Appendix B

**Table A1.** List of persons interviewed during the expert interviews.

| Interview Number (IN) | Name [1] | Function |
|---|---|---|
| 1 | Anonymized | Managing Director of a German inland shipping company |
| 2 | Christian Kunz | Managing Director of the German Federation for the Environment and Nature Conservation (BUND) in Saxony-Anhalt |
| 3 | Anonymized | Coordinator for the implementation of the Water Framework Directive in a river basin unit |
| 4 | Dr. Stefan Cuypers | Managing Director at the Association for Industry–Water–Environmental Protection e.V. (IWU) in Dueren |
| 5 | Tim Rospunt | Head of the Upper Waters Division at the Lower Saxony State Office for Water Management, Coastal Protection and Nature Conservation (NLWKN), Lueneburg Operating Office |
| 6 | Anonymized | Executive position at a water management unit |
| 7 | Tobias Gierra | Project group leader for the Overall Strategy for the Elbe (GKE) at the Elbe Waterways and Shipping Authority (WSA) |
| 8 | Harald Rohr | Vice President (Resort: Water Management) at the Fishing Association (LAV) of Saxony-Anhalt e.V. |
| 9 | Prof. Dr.-Ing. Irene Slavik | Professorship for Sanitary Water Management—Focus on Water Supply at the Magdeburg-Stendal University of Applied Sciences |
| 10 | Dr. Christof Homann | Division Manager for Water Management Information at the Water Association Eifel-Rur (WVER) |

[1] For all interview partners mentioned by name, a signed declaration of consent in accordance with data protection law is available; all other interview partners have agreed to anonymized processing of the data.

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
