# Peer review of "Driving Forces and Socio-Economic Impacts of Low-Flow Events in Central Europe: A Literature Review Using DPSIR Criteria"

_sustainability, doi:10.3390/su151310692_

Round 1

Reviewer 1 Report

Review notes

While I can understand the need to keep the literature review manageable, the choice to include “DPSIR” as an “AND” term in the query is a drastic limitation, effectively limiting the review to what’s been produced by others using that particular framework. When I entered the authors’ search terms into google scholar, I got 49 search results, but over 3,000 when I excluded DPSIR.

If the goal of the research is to provide a “base for the future development of a damage cost database for low-flow events” (lines 18-19) it seems likely that restricting the search to analyses that have used the DPSIR framework will miss some events or drivers of interest.

The authors mention that “the socio-economic impacts of low-flow events have hardly been studied by the scientific community so far” on lines 136-138 but this seems contradicted by the extremely restrictive search that the authors used to keep results manageable. Also, on lines 312-313 they cite and rely on comprehensive overviews that have already been done.

On line 286 the authors mention the focus of the study is predominantly on Central Europe but they do not say how they limited the geographic scope in searching or in categorizing search results. The pilot area is referenced again on lines 691-692. The experts interviewed all seem to be from the area in question.

The authors acknowledge the limits of the search methods on lines 672-676 and say that they interviewed experts to fill gaps that may have been left by overly restrictive search terms. But the lit review appears to have been global, while the experts were mainly already connected to the project and its network.

Table 2 does a good job of enumerating the effects of low-flow events, and the associated discussion is rich with detail and practical considerations. They correctly in my opinion identify sectors where impacts of low-flow events are easier or harder to quantify and to assign a monetary value. It appears that this table is global in scope, but please specify.

I recommend less emphasis on the methodology of the lit review and more on the combined findings from the interviews. Although the end result is informative, it isn’t necessarily new and may not be definitive due to the methodological shortcomings of the lit review. There may also be a spatial mis-match between the lit review and the expert interviews. The relatively timely list of low-flow impacts assembled from pre-existing comprehensive lists and through conversations with experts is probably the most useful finding, but please state the extent to which it applies globally, only within the developed world, or specifically to Central Europe.  

Author Response

Dear Reviewer,

thank you for your revision comments. We have tried to respond to the comments and explain in the attached document how we have implemented the individual items.

Kind Regards

Reviewer 2 Report

In this review article, the authors summarized the drivers and impacts of low-flow events on different socio-economic sectors in Central Europe by exploring the literature. In general, the article is logically clear and well presented. However, there are some minor aspects that need to be modified:

1)    Indeed, the dry summers of 2018 and 2019 caused comprehensive/typical low-flow events in Central Europe, but they are not the only typical low-flow events in history. Is it possible to summarize how many typical low-flow events/years occurs in Central Europe in the 20th and 21st century? Furthermore, what can we learn from those events?

2)    Section 3.4, the authors should focus on the socio-economic impacts of Low-Flow events, rather than drought. Drought is an extended concept and some contents are not related to this paper.

3)    Line 351-364, 370-374, nonstandard use of punctuation marks.

4)    Figure 4 is poorly presented. There is a display of different color depths. Perhaps the author wanted to express the relationship between primary and secondary factors, which should be explained and presented in another way.

5)    In Table 2, is there a mismatch between "sector" and "associated effects"?  "lowering of water levels in rivers and canals make navigation difficult or impossible" partly corresponds to "inland navigation" and partly corresponds to "tourism & recreation ".

6)    Line 328, full name for "WHO" in the first appearance.

      7)    Line 521-522, full names for "EU" and "EC" in the first appearance.

Author Response

(The authors gave the same response as above.)
